# Notch Signaling Molecules as Prognostic Biomarkers for Acute Myeloid Leukemia

**DOI:** 10.3390/cancers11121958

**Published:** 2019-12-06

**Authors:** Paul Takam Kamga, Giada Dal Collo, Federica Resci, Riccardo Bazzoni, Angela Mercuri, Francesca Maria Quaglia, Ilaria Tanasi, Pietro Delfino, Carlo Visco, Massimiliano Bonifacio, Mauro Krampera

**Affiliations:** 1Section of Hematology, Stem Cell Research Laboratory, Department of Medicine, University of Verona, Policlinico G.B. Rossi., P.le L. Scuro, 10, 37134 Verona, Italy; takam.paul@gmail.com (P.T.K.); giada.dalcollo@gmail.com (G.D.C.); federicaresci@hotmail.it (F.R.); ric.bazzoni@gmail.com (R.B.); mercuri.angela83@gmail.com (A.M.); francescamaria.quaglia@gmail.com (F.M.Q.); ilaria_tanasi@hotmail.com (I.T.); carlo.visco@univr.it (C.V.); massimiliano.bonifacio@univr.it (M.B.); 2EA4340-BCOH, Biomarker in Cancerology and Onco-Haematology, UVSQ, Université Paris Saclay, 92100 Boulogne-Billancourt, France; 3Department of Diagnostics and Public Health, University and Hospital Trust of Verona, 37134 Verona, Italy; delfino.pietro@gmail.com

**Keywords:** Notch signaling, AML, biomarkers

## Abstract

The role of Notch signaling in acute myeloid leukemia (AML) is still under investigation. We have previously shown that high levels of Notch receptors and ligands could interfere with drug response. In this study, the protein expression of 79 AML blast samples collected from newly diagnosed patients was examined through flow cytometry. Gamma-secretase inhibitors were used in AML mouse xenograft models to evaluate the contribution of Notch pharmacological inhibition to mouse survival. We used univariate analysis for testing the correlation and/or association between protein expression and well-known prognostics markers. All the four receptors (Notch1–4) and some ligands (Jagged2, DLL-3) were highly expressed in less mature subtypes (M0–M1). Notch3, Notch4, and Jagged2 were overexpressed in an adverse cytogenetic risk group compared to good cytogenetic risk patients. Chi-square analysis revealed a positive association between the complete remission rate after induction therapy and weak expression of Notch2 and Notch3. We also found an association between low levels of Notch4 and Jagged2 and three-year remission following allogeneic stem cell transplantation (HSCT). Accordingly, Kaplan–Meier analysis showed improved OS for patients lacking significant expression of Notch4, Jagged2, and DLL3. In vivo experiments in an AML mouse model highlighted both improved survival and a significant reduction of leukemia cell burden in the bone marrow of mice treated with the combination of Notch pan-inhibitors (GSIs) plus chemotherapy (Ara-C). Our results suggest that Notch can be useful as a prognostic marker and therapeutic target in AML.

## 1. Introduction

Acute myeloid leukemia (AML) is the most common malignant myeloid disorder in adulthood, still incurable in most patients, with a 35–40% five-year overall survival (OS) rate for non-promyelocytic AML [1,2]. The disease is highly heterogeneous both amongst patients and in the same patient, with the occurrence of many blast clones in samples from a single patient [3]. AML is, therefore, a major challenge for personalized medicine. Thanks to the association of advanced diagnostic procedures, including genome-wide molecular-profiling, immunophenotyping, cytogenetic, and clinical features, many biomarkers guiding therapeutic strategy have been defined [2,4,5]. Despite the decisive contribution of currents prognostic markers, all these combining approaches fail to tackle disease complexity and outcome [2,6,7].

Notch signaling is a developmental pathway consisting of four receptors (Notch1–4) and five ligands, including Jagged1-2, Delta-like ligand 1, 3–4 (DLL-1, DLL3–4). The interaction between ligands and receptors induces the release of the intracellular part of the receptors, which acts in combination with transcriptional co-activators and induces expression of target genes [8]. The activation of the pathway has been first described in T-cell acute lymphoblastic leukemia (T-ALL), where more than 50% of patients present hyperactivating mutation of Notch1 genes [9]. Notch components have emerged as prognosis markers in T-ALL as well as in other hematologic malignancies, such as B-cell chronic lymphocytic leukemia (B-CLL) [9,10,11].

In AML, even though the pathway is highly represented, no mutation of the pathway has been described [12]. Nevertheless, some studies have already provided evidence that Notch pathway expression can be correlated with AML patient outcomes [13,14,15,16]. Xu et al. demonstrated that high levels of Notch1, Jagged1, and DLL-4 are associated with poor prognosis [13]. Consistently, Zhang and colleagues used both real-time PCR and Western blot analysis for highlighting the association amongst Notch1, DLL-4, and poor patient survival [14]. The aforementioned studies addressed the role of single receptors and one or two ligands, while growing evidence supports the role of other components of the pathways in hematological malignancies and solid cancer.

In this study, we used flow cytometry to analyze the expression of Notch receptors and ligands on the cell membrane of AML blast cells. We observed higher levels of the four surface Notch receptors of blast cells as compared to CD34+ hematopoietic stem cells. We analyzed the relationship between the expression level of each protein and AML clinical and pathological parameters, including risk stratification, OS, and response to therapy.

## 2. Results

### 2.1. Flow Cytometric Analysis of Membrane Expression of Notch Molecules

To determine whether Notch expression levels at the cell membrane of primary blast cells collected from AML patients at diagnosis were associated with patient outcome, we analyzed the Notch signaling expression in 79 AML samples (Table 1) and 18 CD34+ cell samples from healthy donors. Flow cytometric analysis of cells labeled with fluorochrome-conjugated Notch antibodies showed that Notch1, Notch2, Notch3, Notch4, Jagged1, Jagged 2, DLL-1, DLL-3, and DLL-4were expressed in 86% (59/69), 87% (60/69),58% (40/69), 42% (29/53), 86% (68/79), 32% (22/69), 6% (4/69), 97% (67/69), and 3% (2/53) of patient samples, respectively. The expression levels of Notch1, Notch2, and DLL-3 were higher (*p* < 0.001, *p* < 0.05, *p* < 0.001 respectively) in leukemic cells as compared to CD34+ cells from healthy donors (Figure 1). Except for two donors, Notch3 and Notch4 were expressed only in blast cells.

### 2.2. Correlation between Notch Expression and Known Prognostic Markers

We then analyzed the correlation between Notch protein expression and patient prognostics markers. As DLL-1 and DLL-4 were poorly expressed at the membrane surface of cells from both healthy donors and patients, they were not included in the subsequent analyses. Patient age, gender [17], hemogram, the French–American–British (FAB) subtype, and cytogenetics were considered in the study. Correlation of molecule expression levels with the gender (males vs. females) was carried out in all patients, but only 58 patients (Appendix A) were included in all the analyses. Twenty-one patients were excluded due to insufficient clinical and molecular information. We observed no differences in protein expression levels according to gender (Figure 2A) and age (data not shown). Spearman correlation supported a negative association of white blood cell (WBC) counts with Notch1 (r = −0.31; *p* = 0.012), and Notch2 (r = −0.24; *p* = 0.039) levels, as well as of Hemoglobin (Hb) and Jagged2 (r = −024; *p* = 0.038) (Table 2). Positive associations were found between Hb counts and Jagged1 (r = 0.27; *p* = 0.023) as well as platelet (PLT) counts and Notch1 (r = 0.29; *p* = 0.0166) (Table 2). Considering the FAB leukemic subtype, all four receptors were overexpressed in less mature subtypes (M0–M1), while the expression levels of all receptors and ligands was homogeneous in all the other FAB subtypes (Figure 2B). According to the recommendation of the European Leukemia Network (ELN), we divided the patients into three cytogenetic risk categories: good, intermediate, adverse. Then, we analyzed Notch expression in each group. Patients classified in the adverse group expressed higher levels of Notch3, Notch4, and Jagged 2 as compared to the remaining patients, particularly those classified in the good risk group (Figure 3). Notch1 and Notch2 were also overexpressed in the intermediate risk group, but differences were not statically different.

### 2.3. Correlation with Treatment Outcome and Patient Survival

As the highest levels of Notch3, Notch4, and Jagged2 were found in the high-risk patient group, we asked whether the expression of receptors and ligands was associated with treatment outcome and patient survival. We focused our attention on (i) patient response to induction chemotherapy, (ii) patient outcomes following allogeneic stem cell transplantation (HSCT), (iii) patients’OS. Induction chemotherapy consisted either in 7+3 (Idarubucin + chemotherapy (Ara-C), n = 26) or MICE (Etoposide + Mitoxantrone + Ara-C, n = 25) or other schedules (n = 4). Patients were followed for 36 months. Considering all the patients together, regardless of the type of induction therapy, we observed that blast cells from responsive patients (complete response) displayed higher levels of Notch2 compared to refractory patients (Figure 4A). Chi-square analysis was applied to examine the association between blast cell Notch levels at diagnosis and the outcome of induction therapy. To this aim, patients were classified into 2 groups corresponding to patients with protein expression levels below the overall median of expression (‘low’ group) or above the overall median of expression (‘high’ group), respectively. As far asNotch3, Notch 4, and Jagged2 are concerned, the difference between the two groups was clear-cut (no expression versus significant expression), while for the other receptors and ligands, the expression degree was variable but measurable. Thus, Chi-square analysis revealed a positive association between complete response and low expression of Notch 2 (r = 2.74; *p* = 0.049) and Notch 3 (r = 0.032; *p* = 3.42) (Table 3). Next, we investigated the relation between Notch expression level at diagnosis and outcome following HSCT, which was carried out either after consolidation, or in place of consolidation, or as salvage therapy. Patients’ outcome following HSCT (29 patients) was evaluated after 3 years: 10 patients (34%) were still in complete remission, while 19 patients (66%) relapsed. Blast cells from relapsed patients displayed higher levels of Notch4 (*p* < 0.05) and Jagged2 (*p* < 0.001) as compared to responsive patients (Figure 4B). Consistently, Chi-square analysis confirmed the association between Notch4 expression (r = 2.65; *p* = 0.05), Jagged2 (r = 7.64; *p* = 0.003) and outcome following HSCT (Table 4). We next used Kaplan–Meier analysis, followed by the Log-Mantel Cox-test to evaluate progression-free survival (PFS) and OS of patients according to Notch expression level on blast cells. Longer PFS were observed for patients with blast cells displaying lower levels of Notch3 (Hazard Ratio, HR = 1.441; 95% Confidence interval (CI) (0.479–4.333), Notch4 (HR = 2.969; 95% CI (0.5248–16.80), Jagged1(HR = 2,14; 95% CI (0.740–6.22)) and Jagged2 (HR = 1,415; 95% CI (0.4607–4.346)), and DLL-3 (HR = 0,765; 95% CI (0.1236–4.736)). However, the Log-mantel Cox-test showed that these trends were not significant (Figure 5B). Concerning OS, we observed a significant longer OS for patients with blast cells displaying lower levels of Notch4 (HR = 4.1; 95% CL (1.386–12.13); *p* = 0.011), Jagged2 (HR = 3.5; 95% CI (1.16–10.6); *p* = 0.027),and DLL-3 (HR = 3.5; 95% CL (1.079–11.36); *p* = 0.037)(Figure 5B). Multivariate analysis, based on K-means clustering via principal component analysis (PCA), followed by survival analysis revealed that patients with low expression of Jagged2, Notch3, and Notch4 (Cluster 3 and 4) presented significantly longer survival compared to other patients (Figure 6).

### 2.4. Notch Inhibition Prolongs Survival of AML Mouse Xenograft Model

Our findings suggested that higher levels of Notch proteins affected patients’ survival. Therefore, we tested whether pharmacological inhibition of the Notch pathway in a mouse xenograft model of AML could result in an anti-leukemic effect and, therefore, survival improvement of transplanted mice. Mice were treated as described in Materials and Methods. Analysis of mouse bone marrow at 3 weeks showed that Notch pan-inhibitors (GSIs) alone (GSI-IX and GSI-XII), at the concentrations used in this study, were not able to change the levels of human CD45+ cells (hCD45+) in mouse bone marrow, while Ara-C alone dramatically reduced leukemic burden (Figure 7A,B). The combination regiment of GSIs plus Ara-C significantly reduced the levels of leukemic cells in mouse bone marrow as compared to Ara-C alone (Figure 7B). Interestingly, the OS of mice treated with GSI-IX (20 days) or GSI-XII (21 days) alone was longer than control mice treated with DMSO (16 days) (Figure 7C). Ara-C alone significantly improved mouse survival (23 days), while the combination regiments (Ara-C+GSIs) revealed a survival advantage compared to Ara-C treatment (Ara-C+GSI-IX: 26 days; Ara-C+GSI-XII: 26 days) (Figure 7D).

## 3. Discussion

Identification of accurate biomarkers predicting the response to chemotherapy or the outcome following HSCT, even in patients who failed existing standard-of-care treatments, is still an unmet need in AML. In particular, prognostic and predictive molecular surrogates enabling the selection of patients who are likely to benefit from alternative treatments are not available for clinical testing yet [18].

Our findings highlighted that expression levels of Notch receptors and ligands, as analyzed by flow cytometry, are associated with patient cytogenetic risk stratification, response to induction therapy, outcome following HSCT, and survival. There is currently no agreement on whether the Notch pathway plays a role in AML as either oncogene or tumor suppressor. Nevertheless, the significant expression of Notch components in AML cells is broadly recognized, thus questioning their pathophysiological role and correlation with disease features [19,20,21,22]. Previous studies on single receptors and/or ligands, such as Notch1 and Jagged1, have provided evidence that high levels of Notch components are associated with poor prognosis in AML [13,14,16]. Taking into consideration the redundant effects of the Notch pathway in many pathological processes [8,23,24,25], our study analyzed the prognostic value of the surface expression of each of the nine Notch proteins on AML blast cells. In this analysis, we did not consider the endogenous expression and activation status, as the pathway is poorly activated in AML except when blast cells are in contact with stromal components [26]. Flow cytometric analysis revealed a robust expression in AML cell samples of all Notch receptors and ligands but DLL-1 and DLL-4. This finding is likely due to the lack of protein localization rather than to the lack of protein expression, as PCR, Western blot analysis, and immunohistochemistry have previously demonstrated DLL1 and DLL4 expression in AML samples [13,14].

In T-ALL and B-CLL, Notch pathway overexpression is a consequence of activating mutations and is associated with patient prognosis [9,10]. However, mutations affecting the genes coding for Notch components are rare events in AML as well as in B-ALL [12]. Our group has recently shown that Notch immunophenotypic pattern may predict drug response in B-ALL, suggesting that expression levels of Notch components could be more relevant for disease than mutations [11,27]. This finding could be explained by the interaction between Notch signaling and some protein products of genes that are often mutated in AML, such as *FLT3* [26], *NPM1* [27], and *C/EBPα* [28].

Recurrent mutations are used to define the cytogenetic risk stratification according to ELN classification. We show here the association between Notch expression and cytogenetic risk, by demonstrating the overexpression of Notch3, Notch 4, and Jagged 2 in cell samples from AML patients belonging to adverse cytogenetic groups, while these molecules were lacking in control cells, such as CD34+ cells and peripheral blood mononuclear cells (data not shown). We consistently observed longer OS in high expressers of Notch3, Notch4, Jagged2, and DLL3. Considering Jagged1, a previous report suggested that even though the expression levels of the protein do not differ according to ELN groups, low expression of Jagged1 was associated with significantly shorter OS. However, this previous report was based on the high treated patients, differing from the current study, which includes all patients regardless of the type of therapy. Notch1 and Notch2, although expressed in most of the AML cell samples, were not significantly associated with any of the ELN cytogenetic risk groups. Previous reports suggesting that Notch1 overexpression was a negative prognostic marker in AML were focused on the active forms of the receptors, which are expressed only in a minority of patients [12,17]. Notch1 is far beyond the most studied Notch receptors [8]. Further evidence provided the rationale for also studying other members of the Notch family [19,25]. For instance, mutations in the *NOTCH3* gene have been found in T-ALL, where the Notch1 gene plays a pivotal pathogenetic role [28]; the consequent deregulated expression of the Notch3 receptor supports leukemic cell growth and chemoresistance [29]. Aberrant activity of the Notch3 receptor was also found in other human malignancies, such as pancreatic adenocarcinoma and ovarian cancer, where its expression was associated with poor survival [30]. Similar results were obtained for the Notch4 receptor in epithelial cancers [31,32,33].

Notch deregulation and expression in leukemia could be a consequence of the paracrine crosstalk among Notch ligands and receptors expressed by stromal niche components [11]. In fact, we have previously observed that many Notch molecules are highly expressed and activated in stromal cells from leukemia patients as compared to their normal counterparts. These previous findings suggested that high expression levels of Notch signaling in bone marrow cells go in parallel with persistent modifications of the bone marrow niche, eventually supporting leukemia cell growth and drug resistance [17,18,31]. Similarly, the bone marrow niche is also critical for successful cell engraftment and immunological chimerism following HSCT. Considering that the Notch pathway plays a central role in T cell development, maintenance, and activation [34,35], some transplant-related phenomena, such as graft-versus-leukemia (GvL), could be affected by the expression levels of the Notch components in bone marrow. Accordingly, our findings showed the association between three-year remission following HSCT and the lack of expression of Notch4 and Jagged2 by AML blast cells at diagnosis.

The prognostic value of Notch pathway overexpression is strengthened by the evidence that abrogation of Notch3, Notch4, and Jagged2 signaling increases AML cell chemosensitivity in vitro [17]. In the current study, we also demonstrated in vivo the anti-leukemic effect of Notch inhibition by showing in the mouse xenograft model of AML that GSIs affect leukemic cell expansion, thus prolonging mouse survival. This observation suggests that the poor OS of patients with Notch4, Jagged 2, and DLL-3 cell overexpression at diagnosis may be associated with enhanced drug resistance, while the lack of significant expression of Notch proteins correlates with response to induction chemotherapy.

## 4. Materials and Methods

### 4.1. Patients, Samples, and Cell Lines

All cell samples were collected from AML patients and healthy donors after written informed consent, as approved by the Ethics Committee—Azienda Ospedaliera Universitaria Integrata Verona (N. Prog. 1828, May 12, 2010—‘Institution of cell and tissue collection for biomedical research in Onco-Heamatology’). AML blast cells were obtained from bone marrow or peripheral blood samples of patients at diagnosis (Table 1).

### 4.2. Chemicals and Antibodies

The antibodies used for flow cytometry were: mouse IgG2b-FITC, goat IgG-PE, anti-Jagged1-FITC, anti-Dll3-PE (all from R&D System, Minneapolis, MN, USA), mouse IgG2a-PE, mouse IgG1κ-PE, mouse IgG1-Alexa Fluor 488, anti-Notch1-PE, anti-Notch2-PE, anti-Notch3-PE, anti-Notch4-PE, anti-Dll1-PE, anti-Dll4, (all from Biolegend, San Diego, CA, USA) (DAKO). For blast cell identification, we used anti-CD45-VioBlue, anti-CD45-APC-Vio770, anti-CD34-PerCP, and anti-CD117-APC (Miltenyi Biotec, Germany).

### 4.3. Flow Cytometry

AML cells were identified as CD45+, CD34+, CD38− cells by flow cytometry. Cells were initially selected using a morphological gate based on forward scatter (FSC) and side scatter (SSC) parameters. The subsequent evaluation was performed on CD45+ versus SSC and then CD34+ versus CD38− to identify myeloid blasts. A threshold was fixed on FSC to exclude cellular debris. The analysis of CD34 and CD38 expression was performed both in AML cell lines and in primary AML cells from patients. The percentage of CD45+, CD34+, and CD38− was used to evaluate the expression of Notch receptors and ligands compared to isotype-specific antibodies.

### 4.4. Notch Receptor Immunophenotype

Expression of Notch receptors and ligands was analyzed by flow cytometry using phycoerythrin (PE)-conjugated antibodies against Notch receptors and ligands. As previously shown [26], at least 3 × 10^5^ cells were labeled for 15 min in the dark at room temperature, either with the specific conjugated antibodies or their specific isotype. Unbound antibodies were removed by washing the cells with 1X PBS. The phenotype was assessed by FACS Canto II, and data were analyzed by FlowJo software. Results were expressed as the relative median of fluorescence intensity (rMFI), defined as the ratio of specific fluorescence (mean fluorescence of blast cells incubated with the conjugated antibody) over non-specific fluorescence (mean fluorescence of blast cells incubated with the specific conjugated isotypic control). The expression threshold was settled at rMFI = 1.

### 4.5. Xenograft Mouse Model

NOD/Shi-scid/IL-2Rγnull (NOG) mice were purchased from Taconic (Germantown, NY, USA) and kept in pathogen-free conditions in the animal facility at the Interdepartmental Centre of Experimental Research Service (CIRSAL) of the University of Verona, as approved by the Italian Health Ministry (Autorizzazione n°1294/2015-PR). U937 AML cell line (1 × 10^6^) was injected into the tail vein of totally irradiated (1.2 Gy, ^137^Cesium source), 8–12-week-old mice. At day 9 post-injection, mice were assigned to one of the following treatment arms: DMSO or Ara-C (100 mg/kg), all administered through daily intraperitoneal injection for 3 days (from day +9 to +11). On day +12, each treatment arm was split into three arms: GSI-IX (10 mg/kg) or GSI-XII (10 mg/kg) or their vehicle (DMSO). Animals were sacrificed after 3 weeks from cell line injection, and bone marrow leukemic burden was evaluated as the percentage of human CD45+ cells. For survival assays, animals were sacrificed when body weight loss was equal to 20% according to ethical regulation.

### 4.6. Statistical Analysis

Statistical analysis was performed using GraphPad Prism software (La Jolla, CA, USA). Mann–Whitney and Kruskal–Wallis methods were used to compare two groups or more than two groups, respectively. Pearson Chi-square analysis was used to test the association between variables. Survival curves were calculated by the Kaplan–Meier method. Multivariate analysis was performed through PCA analysis and K means clustering, as previously described [36].

## 5. Conclusions

In conclusion, our study suggests that the assessment at diagnosis of Notch signaling protein expression and, mainly Notch3, Notch4, and Jagged2, overexpression in AML blast cells has a prognostic value to predict patient outcome. These findings represent the rationale for larger studies aimed at improving the treatment of AML through Notch-targeted therapies.

## Figures and Tables

**Figure 1 cancers-11-01958-f001:**
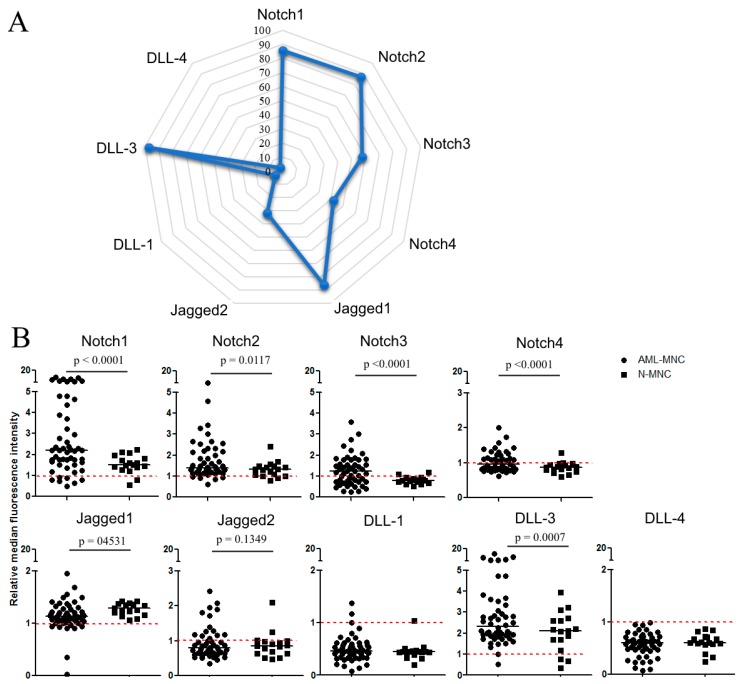
Notch expression in acute myeloid leukemia (AML) blasts. Flow cytometric analysis of AML blasts and CD34+ cells using conjugated antibodies specific to extracellular Notch receptors and ligands. (**A**) Proportion of patients (%) expressing each Notch receptor and ligand. (**B**) Notch expression level in AML blast cells (n = 54) and healthy donor CD34+cells (n = 18). The median of fluorescent intensity for each receptor and ligand was normalized by dividing each value by the median of fluorescent intensity of its corresponding isotype. Mann–Whitney test was used to analyze the differences between the two groups.

**Figure 2 cancers-11-01958-f002:**
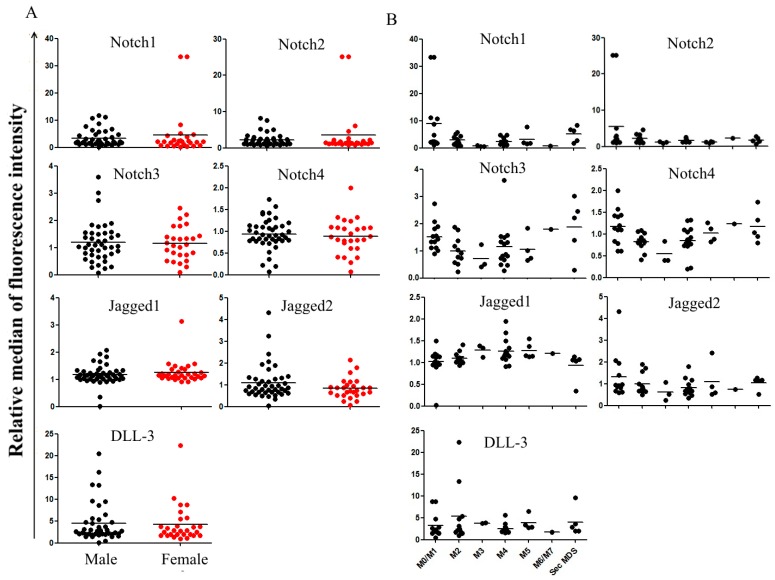
Expression in primary AML samples according to French–American–British (FAB) subtype and gender. Patient samples analyzed for Notch expression were classified according to patient gender (male n = 42; female n = 27) (**A**) and their FAB subtype by flow cytometry (**B**). Data are represented as the relative median of fluorescence intensity for each antibody normalized to specific fluorochrome-conjugated controls.

**Figure 3 cancers-11-01958-f003:**
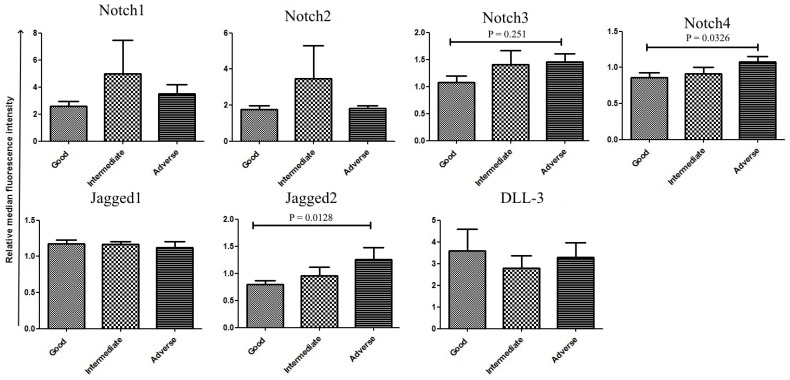
Expression in primary AML samples according to patient stratification. Patient samples analyzed for Notch expression were classified according to the European Leukemia Network (ELN) genetic risk stratification; good (n = 21), intermediated (n = 13), and adverse (n = 18). Mann–Whitney test was used to analyze the differences between the two groups.

**Figure 4 cancers-11-01958-f004:**
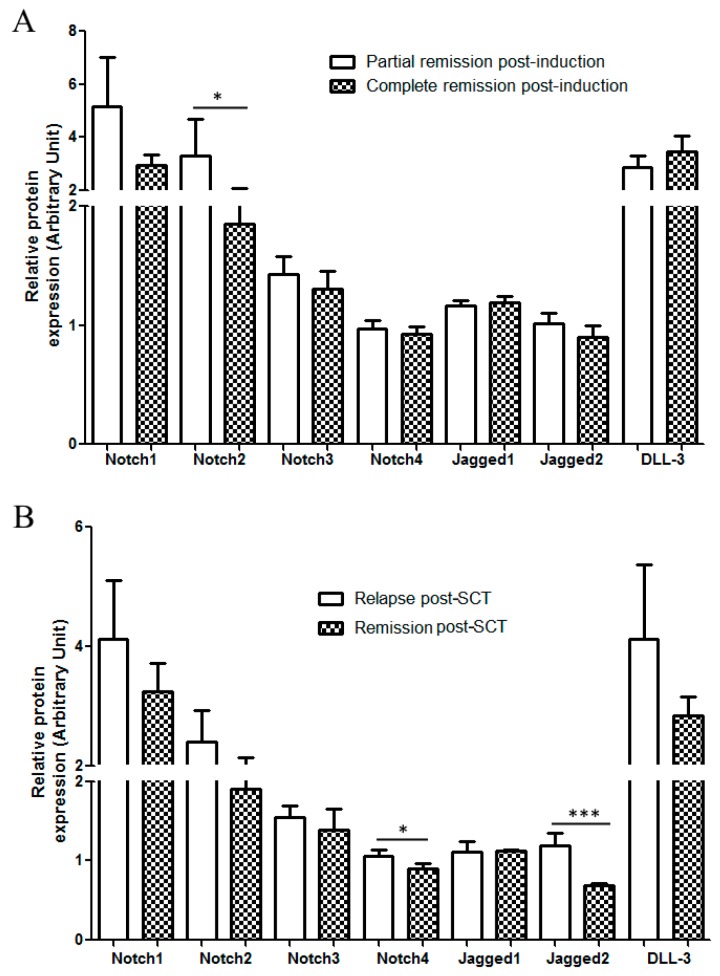
Expression in samples from patients according to their outcome following induction therapy and allogeneic hematopoietic stem cell transplantation (HSCT). Patient samples analyzed for Notch expression were classified according to Notch expression levels (low or high) and response to induction therapy (**A**) or three-year remission after HSCT (**B**). * *p* < 0.05; *** *p* < 0.001.

**Figure 5 cancers-11-01958-f005:**
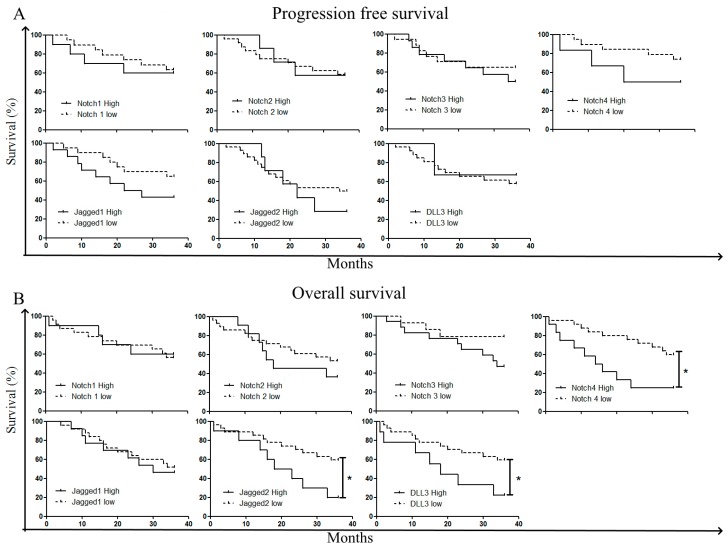
Survival according to Notch expression levels. Patient samples analyzed for Notch expression were classified according to Notch expression levels (low or high). Log-rank (Mantel–Cox) analysis was used for establishing the difference in progression-free survival (PFS) (**A**) and overall survival (OS) (**B**) between the two groups. * *p* < 0.05.

**Figure 6 cancers-11-01958-f006:**
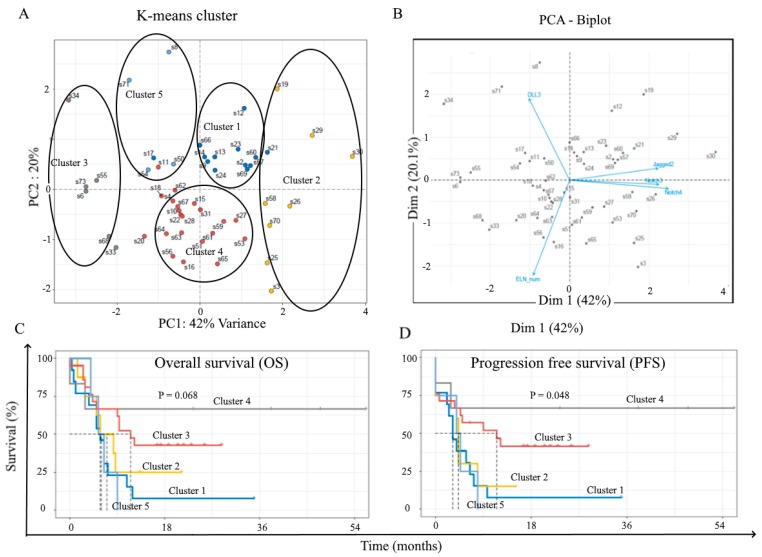
Multivariate data analysis according to ELN groups, Notch3, Notch 4, Jagged 2, and DLL-3 expression levels. Thanks to R software, dimensionality reduction was performed through K-Means Clustering (**A**) and principal component analysis (PCA) (**B**). Log-rank (Mantel–Cox) analysis was used for establishing the difference in overall survival (OS) (**C**) and progression-free survival (PFS) (**D**) between the two groups.

**Figure 7 cancers-11-01958-f007:**
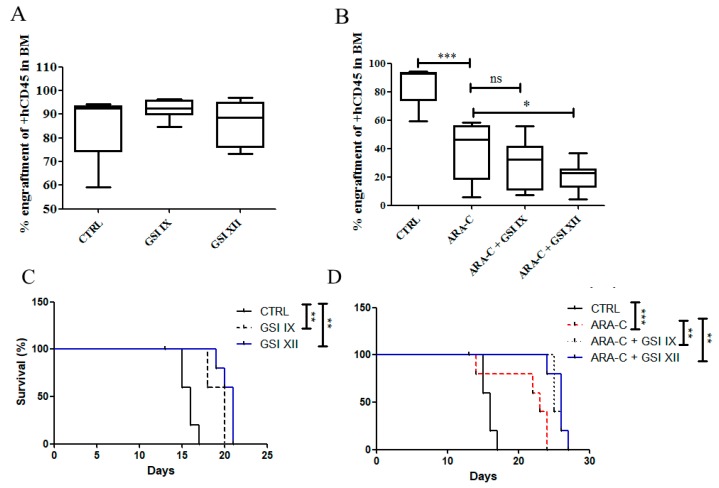
Notch inhibition reduces bone marrow leukemic burden and prolongs the survival of AML mouse models. (**A**,**B**) Flow cytometry analysis of human CD45+ cells (hCD45+) in bone marrow (BM) samples obtained from mice transplanted with AML cell line U937.(**A**) Starting from day +12 post-engraftment, mice were treated for 3 days with either GSI-IX (10 mg/kg) or GSI-XII (10 mg/kg) or their vehicle (DMSO). (**B**) Starting from day +9 post-engraftment, mice were treated for 3 days with Ara-C followed by 3 days of GSI-XII (10 mg/kg) or its specific vehicle (DMSO). The assay was performed with at least 5 mice in each group. Data are reported as mean ± SEM. (**C**,**D**) Survival of mice transplanted with U937 and subdivided into five treatment groups. Treatments were administrated as described above (**A**,**B**). Differences in survival curves were analyzed with the Log-rank (Mantel–Cox) Test. * *p* < 0.05; ** *p* < 0.01; *** *p* < 0.001; ns = not significant.

**Table 1 cancers-11-01958-t001:** Patients characteristics. 7+3 protocol: 3 days of anthracycline + 7 days of chemotherapy (Ara-C); MICE protocol: 3 days of Mitoxantrone and Etoposide + 7 days of Ara-C; FLAI protocol: 5 days of Fludarabine and Ara-C + 2 days of Idarubicin; HSCT: allogeneic hematopoietic stem cell transplantation.

Patients	N = 79
Gender	Females = 31
	Males = 48
Median age	52 (16–74)
FAB Subtypes (n = 57)	M1/M0 = 14
	M2 = 13
	M3 = 3
	M4 = 16
	M5 = 5
	M6/M7 = 1
	Secondary to myelodysplasia = 5
	Other = 1
Hemoglobin	Hb (g/dL) = 8.7 (3.16–12.9)
Platelets	PLT count (109/L) = 67.5 (6–319)
White blood cells	WBC count (109/L) = 20 (0.5–21.8)
ELN stratification (n = 51)	Good = 22
	Intermediate = 17
	Adverse =18
Induction therapy (58)	7+3 = 26
	MICE = 26
	FLAI = 2
	Other = 4
HSCT (n = 29)	Three year remission = 10
	Relapse within 3 years = 19

**Table 2 cancers-11-01958-t002:** Correlation between hemogram and Notch signaling expression in acute myeloid leukemia (AML) blast cells. Spearman analysis was used to establish a correlation between levels of Notch in blast cells and parameters of the hemogram, including(**A**) white blood cells (WBC)**,** (**B**) Hemoglobin (Hb), (**C**) platelets (PLT). * *p* < 0.05; ns: not significant.

**A**	**WBC**	**Notch1**	**Notch2**	**Notch3**	**Notch4**	**Jagged1**	**Jagged2**	**DLL-3**
		54	54	54	54	54	54	53
	r	−0.3051	−0.2424	0.07906	−0.08409	0.2046	−0.1569	0.03657
	*p*-value	0.0124	0.0387	0.2849	0.2727	0.0689	0.1286	0.3974
	Statistics	*	*	ns	ns	ns	ns	ns
**B**	**Hb**	**Notch1**	**Notch2**	**Notch3**	**Notch4**	**Jagged1**	**Jagged2**	**DLL-3**
		54	54	54	54	54	54	53
	r	0.01171	−0.1183	0.03234	0.0008389	0.2720	−0.2440	−0.1473
	*p*-value	0.4665	0.1971	0.4082	0.4976	0.0233	0.0377	0.1462
	Statistics	ns	ns	ns	ns	*	*	ns
**C**	**PLT**	**Notch1**	**Notch2**	**Notch3**	**Notch4**	**Jagged1**	**Jagged2**	**DLL-3**
		54	54	54	54	54	54	53
	r	0.2905	0.02284	0.05585	0.05406	0.1449	0.03023	−0.02843
	*p*-value	0.0166	0.4349	0.3442	0.3489	0.1480	0.4141	0.4199
	Statistics	*	ns	ns	ns	ns	ns	ns

**Table 3 cancers-11-01958-t003:** Association between Notch expression levels and response to induction treatment. Patient samples analyzed for Notch expression were classified according to Notch expression levels (high expression versus low expression) and remission post-induction therapy. Chi-square analysis was used to test the association between low protein levels and response to induction therapy. * *p* < 0.05; ns: not significant.

	Notch1	Notch2	Notch3	Notch4	Jagged1	Jagged2	DLL-3
Chi-score	0.1822	2.737	3.424	0.01225	0.3810	2.915	0.1822
*p*-value	0.3348	0.0490	0.0321	0.4559	0.2685	0.0878	0.3348
Statistics	ns	*	*	ns	ns	ns	ns

**Table 4 cancers-11-01958-t004:** Association between Notch expression levels and three-year remission post-HSCT. Patient samples were classified according to Notch expression levels (High expression versus Low expression) and three-year remission post-HSCT. Chi-square analysis was used to test association between low protein levels and response to induction therapy. * *p* < 0.05; ns: not significant.

	Notch1	Notch2	Notch3	Notch4	Jagged1	Jagged2	DLL-3
Chi-score	0.03	0.69	1.15	2.65	0.69	7.64	1.01
*p*-Value	0.43	0.20	0.14	0.05	0.20	0.003	0.15
Statistics	ns	ns	ns	*	ns	*	ns

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
