# Peer review of "Notch Signaling Molecules as Prognostic Biomarkers for Acute Myeloid Leukemia"

_cancers, 2019, doi:10.3390/cancers11121958_

Round 1

Reviewer 1 Report

Dear Authors,

I appreciate your work, however would you answer my concerns;

1. In Figure1 - would you calculate the significance p value for jagged2, DLL-1, DLL-3 and DLL-4? it seems DLL-3 expression is significantly higher. So it is better to write over the significance value for all. 

  2. In Figure3 - Notch1 and Notch2 is highly expressed in intermediated groups as compared to Good and Adverse while Notch4 and Jagged2 is significantly in Adverse higher than Good and intermediated groups and on other-hand DLL3 is higher in Good and Adverse. would you explained in detail about these differences?   3. Higher level of the 4 surface Notch receptors of blast cells as compared to CD34+ hematopoietic stem cells. Did you measure the receptors levels in other cells as well like T cells, B cells? if so, whats the contribution of other cells in this case ? if Notch signaling occur only in blast cells????   4. what are the inter-connecting pathways to Notch signaling which might be trans- modulating this cascade reaction?   5. would you measure the transcription factors involved in this differences ?   Thanks, LR

Reviewer 2 Report

The manuscript is well written and address a relevant topic for biologic characterization, with implication for prognosis and target therapy for AML.

Nevertheless there are some points to be addressed:

AML patients are classified with FAB morphologic classes and must be reclassified with WHO classification.

Mutational status according to ELN 2017 reccomendations must be reported for the patients for which is available

The study does not provide a multivariate analysis of the proposed prognostic factors (expression of Notch1, Notch2, Notch 3, Notch4, Jagged1, Jagged 2, DLL-1, DLL-3 and DLL-4) and other well known prognostic factors (ELN risk groups, karyotype, fusion genes such as CBF and bcr/abl, mutational status of FLT-3, NPM1, p53). This is even most important as 21 cases were excluded from the survival analysis due to incomplete clinical informations

Czemerska et al (ref 15, not present in the text but only in References section), reported that Jagged1 low levels of expression is an adverse prognostic factor. As the Authors reported an opposite finding, these must be discussed

Lobry et al (ref 20,not present in the text but only in References section) and Kannan et al (ref 21, not present in the text but only in References section), report that they found high expression levels of RNAs but not proteins of the the Notch family members. Also these findings must be discussed, being apparently in contrast with the findings reported by the Authors relatively to the expressions of the proteins analysed

Round 2

Reviewer 1 Report

Dear Authors,

I am happy with the reply comments.

Thanks,

LR

Reviewer 2 Report

The Authors have answered to all the proposed queries